# Non-Muscle Myosin 2A (NM2A): Structure, Regulation and Function

**DOI:** 10.3390/cells9071590

**Published:** 2020-07-01

**Authors:** Cláudia Brito, Sandra Sousa

**Affiliations:** 1Group of Cell Biology of Bacterial Infections, i3S-Instituto de Investigação e Inovação em Saúde, IBMC, Universidade do Porto, 4200-135 Porto, Portugal; claudia.brito@ibmc.up.pt; 2Programa Doutoral em Biologia Molecular e Celular (MCBiology), Instituto de Ciências Biomédicas Abel Salazar, Universidade do Porto, 4099-002 Porto, Portugal

**Keywords:** non-muscle myosin 2A (NM2A), NM2A activity regulation, NM2A filament assembly, actomyosin cytoskeleton, cell migration, cell adhesion, plasma membrane blebbing

## Abstract

Non-muscle myosin 2A (NM2A) is a motor cytoskeletal enzyme with crucial importance from the early stages of development until adulthood. Due to its capacity to convert chemical energy into force, NM2A powers the contraction of the actomyosin cytoskeleton, required for proper cell division, adhesion and migration, among other cellular functions. Although NM2A has been extensively studied, new findings revealed that a lot remains to be discovered concerning its spatiotemporal regulation in the intracellular environment. In recent years, new functions were attributed to NM2A and its activity was associated to a plethora of illnesses, including neurological disorders and infectious diseases. Here, we provide a concise overview on the current knowledge regarding the structure, the function and the regulation of NM2A. In addition, we recapitulate NM2A-associated diseases and discuss its potential as a therapeutic target.

## 1. Superfamily of Myosins

The cell cytoskeleton is an interconnected and dynamic network of filaments essential for intracellular organization and cell shape maintenance. It provides mechanical support during critical cellular functions, including cell migration and cell division [1]. The cytoskeleton is composed of three main types of polymers: Actin filaments, microtubules and intermediate filaments [2]. Actin filaments and microtubules display molecular and structural polarity, serving as highways for motor proteins such as kinesins, dyneins and myosins, thus ensuring the directionality of intracellular transport of cargos [1,3]. Besides transport functions, the association of motors to actin filaments and microtubules plays an important role in the intracellular cytoskeleton organization [4].

The myosin superfamily includes more than 30 classes of molecular motors distributed in all eukaryotic organisms [5,6,7]. The human genome contains 38 genes encoding myosin molecules which, based on sequence similarities, are divided into 12 classes [8] (Figure 1). Myosins interact with actin and hydrolyze ATP, converting chemical energy into mechanical force to produce tension and propel the sliding of actin filaments in eukaryotic cells [9,10,11]. Myosin-generated forces serve several processes and pathways demanding force and movement such as cell motility, cytokinesis, intracellular trafficking, signal transduction and muscle contraction [9,12,13,14,15,16]. The specific properties of myosin molecules regarding their structure, enzymatic activity and regulation determine the particular cellular functions in which they are involved. Although almost all members of the myosin family share the biochemical ability to undergo a cyclical ATP-dependent binding to actin, they exhibit great variations in the rate and equilibrium constants of the conserved actomyosin ATPase cycle [17]. In addition, some myosins lack an actin-activated ATPase module and act as pseudo-enzymes [18,19].

The myosin superfamily was categorized into four distinct groups considering their differences in the ATPase activity, time spent attached to actin (duty ratio) and velocity with which they slide on or move along actin. These features outline the type of mechanical activity of each myosin category, classified as fast movers, slow/efficient force holders, strain sensors or processive transporters (Box 1) [20].

Regarding their architecture, all the myosins harbor a conserved head domain followed by a neck region, which allows the binding of myosin light chains (MLCs), as well as members of the calmodulin (CaM) family and CaM-related proteins. The C-terminal tail domain shows high variability among the classes of myosins and contributes to specificity towards binding partners, cellular localizations and functions [8,16] (Figure 2A).

Box 1Myosin classification following kinetic parameters.**Processive transporters** myosins spend a long fraction of the kinetic cycle strongly bound to actin (high duty ratios) and can take multiple steps without detaching (**processivity**). Myosin-V and -VI are exemples of processive enzymes that transport cargos such as melanosomes or endosomes along the actin cytoskeleton to their cellular destination. **Strain sensors** such as class I myosins spend just a small fraction of the actin-activated ATPase cycle bound to actin (low duty ratio) and are the interconnectors between the plasma membrane and the cytoskeleton to regulate both cell migration and membrane tension. Members of Myosin-2 subfamily display low duty ratios and thus also function as strain sensors. Moreover, their ability to form bipolar filaments, slide actin bundles past each other and power contractility in cells, make them prototypes of **force holders**. Myosin-2 molecules assembling into large sarcomeric filaments to contract skeletal muscle are considered **fast movers**.

## 2. Myosin-2 Subfamily

The myosin-2 subfamily is the largest class of myosins, whose members are referred as “conventional myosins” [21], and include skeletal, cardiac and smooth muscle myosins, as well as non-muscle myosin-2 (NM2) isoforms [9]. Besides being highly expressed in non-muscle cells, NM2s are also present in muscle tissues where they play roles in tension maintenance and myofibrillogenesis [22,23]. In non-muscle cells, NM2s associate with actin filaments into distinct actomyosin structures, which are very dynamic and self-reorganize according to the cellular needs [12,24]. This plasticity sustains the key role of NM2s in cell division, migration and other cellular processes that rely on rapid actomyosin remodeling.

## 3. Structure and Characterization of NM2

NM2 molecules are hexameric complexes composed of three pairs of polypeptides: Two non-muscle heavy chains (NMHC2s, 230 kDa), two regulatory light chains (RLCs, 20 kDa) and two essential light chains (ELCs, 17 kDa) (Figure 2A). Together, the ELCs and the RLCs are named MLCs. While the genomes of *Dictyostelium discoideum, Saccharomyces cerevisiae and Drosophila melanogaster* harbor a single gene encoding NMHC2 [25,26,27], vertebrates hold three paralog genes (*MYH9*, *MYH10* and *MYH14*), which are located in different chromosomes and encode three NMHC2 isoforms (NMHC2A, 2B and 2C, respectively) [28,29,30,31,32,33]. Regarding their primary sequence, NMHC2s exhibit 60–80% identity [34,35], showing the highest homology in the motor domains and increased diversity in the tails. The NMHC2s together with the MLCs are termed accordingly NM2A, NM2B and NM2C [36]. In cells, the variety of NM2 molecules is further increased by alternative splicing [28,37,38]. Despite their high conservation at the sequence and structural level, NM2 isoforms display significant differences regarding the dynamics of the myosin filament assembly, the ATPase activities and the duty ratios [21,28,39]. In addition, each isoform displays differential tissue/organ expression and intracellular distribution [37,38,40,41,42,43]. Nevertheless, they also have redundant and interchangeable functions, as demonstrated by the fact that all NM2 isoforms are able to proficiently support cytokinesis [44].

In its N-terminus, the heavy chains contain the conserved motor domain, where the ATP and actin binding sites are located. The neck region consists of an extended helix with two IQ motifs where the MLCs bind non-covalently. Phosphorylation of RLC Serine 19 (Ser19) stimulates the actin-activated ATPase activity and promotes filament assembly [34,45]. The ELCs are important stabilizers of the NMHC structure. In humans, MLCs are encoded by a wide set of genes: Three of them encode for RLCs (*MYL9*, *MYL12A* and *MYL12B*) and two for ELCs (*MYL6* and *MYL6B*) [46]. *MYL9* and *MYL6* may undergo alternative splicing, which combined to the existent five MLCs and three NMHCs increases the variety of NM2 multimeric complexes. While RLCs are abundantly expressed in tissues and interact with all NMHC paralogs, ELC MYL6 only interacts with NMHC2C [47,48]. This suggests that at the cellular level, MYL6 might specifically regulate NMHC2C in space and time, possibly assigning specific functions to NM2C that go beyond the mechanical and kinetic differences between the three NM2 isoforms [21]. NM2C is phylogenetically as related to NM2A and NM2B as to smooth muscle myosin (SMMHC, Figure 1) [28,49], which, interestingly, only binds to ELC MYL6 as well [47,48]. How NMHC2C recognizes ELC MYL6 and the impact of this specific interaction on NM2C function remains to be uncovered.

The C-terminal region of NMHCs constitutes the tail, which is particularly important for the proper subcellular localization of the different NM2 isoforms. In contrast to the conserved motor domain, the tail is variable and unique to each myosin, determining specific functions in cells. The C-terminal α-helical rod domain is a long (~1100 amino acids) region, responsible for NMHCs homodimerization and formation of the coiled-coil tails on the NM2 units [42,50] (Figure 2A). Whenever RLC is unphosphorylated, the motor domains and the tails directly interact, generating an inactive compact structure (Figure 2B). Activation occurs upon phosphorylation on RLC Ser 19, mediated by the calcium-calmodulin-Myosin light chain kinase (MLCK) pathway [51] (Figure 2B). In their active conformation, NM2 tails interact antiparallelly and self-associate into ~300 nm long bipolar filaments (Figure 3A) that contain on average 30 NM2 molecules [21,34]. These bipolar filaments are the working units that crosslink and/or push actin filaments past each other, creating different meshworks of actomyosin bundles such as stress fibers (Figure 3B).

## 4. Assembly of NM2A Filaments

NM2 bipolar filaments have been long considered as homotypic polymers. However, recent studies demonstrated that NM2A molecules co-assemble both in vivo and in vitro either with NM2B or NM2C, forming heterotypic filaments (Figure 3C) [52,53,54]. In addition, the co-assembly of NM2 isoforms with the pseudoenzyme Myosin 18A (Myo18A) was also demonstrated (Figure 3C) [55]. This suggests that cells may adapt the composition of the filaments to control the dynamics of the actomyosin cytoskeleton and to exert more complex functions [56,57,58]. Heterotypic myosin polymers have been the focus of recent studies as they potentially represent a new layer of spatiotemporal regulation of NM2.

NM2A/NM2B co-polymers assembled at the leading edge of migrating cells were proposed to cooperate and facilitate cell motility. Due to their different disassembly rates, NM2A and NM2B isoforms might be self-sorted to different localizations during retrograde flow, to support cell polarization required for motility [54]. Furthermore, the assembly of NM2A/NM2B filaments was suggested to modulate the processive capacity of the NM2 polymers. Compared with NM2B, NM2A molecules have lower duty ratios, suggesting that NM2A homotypic filaments display a non-processive movement. Yet, NM2A/NM2B co-filaments move processively in vitro, on a viscous environment resembling the intracellular milieu, depending on the ratio of the two paralogs [59]. Whether NM2A can act as a processive myosin in vivo remains elusive.

Regarding the co-polymerization of NM2 and Myo18A, it was hypothesized that Myo18A—which does not self-assemble, lacks the ATPase domain and possesses extra N- and C-terminal regions—assembles with NM2 isoforms to control their assembly properties, localization and interaction with binding partners, rather than to contribute to contraction [55]. In particular, in its N-terminus, Myo18A contains a PDZ domain, a protein–protein interaction region [60] that may recruit additional regulatory proteins to the bipolar filament.

These findings paved the way to uncover the role of heterotypic filaments and to understand why and how they assemble.

NM2 units also organize into higher-order superstructures called NM2 stacks, which slowly self-organize and remodel the cytoskeleton [56,57]. NM2 stacks contain several NM2 filaments parallel to an actin bundle, resembling muscle sarcomere structures. Their assembly depends on the ATPase function of NM2, a dynamic actin cytoskeleton, and actin-regulating molecules [56]. Recent findings revealed that Myosin-18B promotes the assembly of NM2 stacks for the maturation of contractile actomyosin fibers [61]. Although NM2 stacks were described both in vitro and in vivo [58,59], the exact mechanisms of stack assembly as well as their function are unknown.

## 5. Kinetic and Mechanical Properties of NM2A

Actin contractile movement depends on the mechanochemical cycle of NM2 molecules, through which they convert chemical energy into mechanical force. In the absence of ATP, myosin is maintained in the so-called rigor state, in which the head domains are tightly attached to actin filaments. ATP binding to NM2 triggers conformational changes that induce the detachment of myosin from actin. The myosin ATPase domain hydrolyses ATP into ADP and phosphate (Pi) whilst the neck domain goes from a curved to a straight conformation, a step that primes the re-attachment of myosin to the actin. Simultaneously to the Pi release, the neck rotates and produces the contraction of the actin filament in a process called powerstroke. The ADP molecule is further released from the actin-bound myosin and the cycle restarts (Figure 4).

The NM2 subfamily comprises the slowest myosins regarding their enzymatic activity and velocity to translocate actin filaments [62,63]. While the enzymatic cycle is conserved for all NM2 paralogs, their kinetic and mechanical properties are variable [45,63] due to subtle differences on the primary sequence of the motor domains. NM2A propels actin filaments two–three times faster than NM2B or 2C [41,63,64] and shows the lowest duty ratio [50]. NM2A-enriched filaments more likely detach from actin and undergo higher rates of turnover and recycling [54], which is supported by the fact that NM2A is mainly found in regions of active actin remodeling. Thus, NM2A fits better in contractile processes including movement and reorganization of the actin cytoskeleton. NM2B is significantly slower, spending more time bound to actin, and therefore better fits tasks that require tension maintenance. In agreement, NM2B is more prominent in stress fibers where it exerts tension [52]. Yet, one should not forget that NM2 filaments can be heterotypic and subunit composition must be considered to fully understand their function and cellular location.

## 6. NM2A Regulation

NM2 isoforms are primarily regulated by phosphorylation on Ser19 of the RLC via calcium-calmodulin-dependent MLCK or Rho-associated protein kinase (ROCK) [51,66]. This phosphorylation releases NM2 from its inactive conformation (Figure 2B), promoting the assembly of myosin filaments and increasing by 1000-fold the ATPase activity [51,66]. In addition to Ser19, Threonine 18 (Thr18) also gets phosphorylated, further increasing ATPase activity [67,68]. In contrast, phosphorylations mediated by Protein kinase C (PKC) on RLC Serine 1 (Ser1), Serine 2 (Ser2) and Threonine 9 (Thr9) are inhibitory as they were associated to a decreased rate of ATP hydrolysis in vitro, and stress fiber disassembly and mitotic arrest in cells [69,70]. These phosphorylations allosterically reduce the MLCK-induced Ser19 phosphorylation, thus decreasing the ATPase activity of NM2 [71]. Furthermore, NM2A activity can be downregulated by the phosphatase MYPT1 that dephosphorylates RLC on Ser19 and Thr18 [17,72]. A fine balance between myosin kinases and phosphatases determines the extent of myosin activation at the right place and time (Figure 5). A novel regulatory phosphorylation on RLC Tyrosine 155 (Tyr 155) was recently demonstrated to spatially control the assembly and function of NM2 in migrating cells [73]. This phosphorylation is mediated by epidermal growth factor receptor (EGFR) and impairs the interaction between NMHC2s and RLCs, specifically at migratory cell protrusions, restricting the assembly of functional NM2 molecules [73]. Given that the majority of Tyr kinases are negatively regulated and only activated in very specific conditions [74], it is tempting to speculate that other unexplored NM2 Tyr phosphorylations, either on the light or the heavy chains, further regulate its assembly, localization and function. Of note, the Src-mediated phosphorylation on NMHC2A Tyr 158 was reported in response to bacterial infections [75], however, whether this modification affects NM2A assembly or activity is unknown.

NM2A filament assembly is also modulated by the coiled-coil tail and non-helical tailpiece of NMHC2 isoforms, through phosphorylation events [17,76] and specific interactions with binding partners such as the S100 calcium-binding protein A4 (S100A4) and the lethal giant larvae (Lgl) [77,78,79]. NMHC2 phosphorylation is critical for filament formation and regulation of NM2 in amoeba [17], yet in mammals, the consequences of NMHC2 phosphorylation are still being defined. aPKCζ and casein kinase 2 (CK2), among other kinases, phosphorylate the tail domain [80,81,82], which promotes the disassembly of NM2 bipolar filaments. Mutations mimicking PKC-induced phosphorylations lead to NM2 filament depolymerization and increased migratory velocity in cancer cells [83]. Apart from phosphorylation events, other factors control the NM2 activity and function (Figure 5). This includes the interaction with binding partners and the assembly of NM2 heterotypic filaments. S100A4, a member of the S100 calcium-binding proteins, interacts with the tail, favoring the depolymerization of NM2A filaments [84,85]. Lgl bound to the coiled-coil region of NMHC2A also controls NM2A filament formation and localization [79]. S100-P, another member of the S100, interacts with NMHC2A and regulates NM2A filament dynamics and cell migration [86], through an unknown mechanism. NMHC2 displays different affinities towards distinct actin isoforms, which modulates the NM2 localization into specific areas or cellular structures, as filopodia and stress fibers. This suggests that the tail domain does not solely determine the cellular distribution of NM2. Distinct actin isoforms also fine-tune NM2A ATPase activity, which is 4-fold higher with β- and γ-actin than with skeletal muscle α-actin [87]. In summary, there is a wide range of mechanisms that regulate NM2 and that do not follow a unique pattern but rather depend on the cell spatiotemporal state. Thus, the variety of roles played by NM2 on particular cellular processes rely on multiple layers of regulation.

## 7. NM2A in Development

NM2A expression is crucial during the early stages of mice embryonic development. The use of NM2A knockout mice, which die at embryonic day 6, revealed the key role of NM2A in the formation of functional visceral endoderm [88,89]. The lack of NM2A impairs the formation of E-cadherin-mediated adherent cell–cell junctions, leading to a disorganized endoderm and loss of cell polarization [89]. Moreover, multinucleated cells were observed in NM2A knockout embryos, suggesting cell division defects [40]. Interestingly, the depletion of NMY-2, one of the two NMHCs expressed by *C. elegans*, impairs cytokinesis in the embryos and further development [90]. Similar results were observed when NMY-2 ATPase activity was abolished or partially ablated, demonstrating that the motor activity of myosin is central during cell division [91]. In mice, the expression of the low ATPase activity mutant NM2A-R702 induces defects in placenta formation [92], suggesting that full active NM2A is required for the process. Moreover, when the *MYH9* gene is replaced by *MYH10* or *MYH14*, under the promoter of *MYH9*, mice normally develop visceral endoderm but die before placenta formation due to defects in angiogenesis and cell migration [40].

NM2A plays important roles in development at the tissue level and its conditional ablation or mutation explains mechanistically some of the illnesses linked with the so-called *MYH9*-related disorders (*MYH9*-RD, described below) [93]. For instance, NM2A is involved in the development of kidney tubules [94] and its lack or mutation is often correlated with nephropathies in *MYH9*-RD patients. In addition, some glomerular disorders are related to a dysregulated expression of NM2A [95]. Non-canonical Wnt-planar cell polarity pathways regulate NM2A activity and play an important role in polarization of the auditory epithelium during development [96]. Compromised NM2A activity may drive the loss of the planar asymmetry and mislocalization of junction proteins such as vinculin [96,97]. This can disrupt sound perception and thus be related to deafness, one of the disorders associated with *MYH9*-RD.

## 8. NM2A in Cell Division, Adhesion and Migration

*Cell division:* The use of various models [98,99,100] revealed that NM2A is required whenever a cell divides into two daughter cells [100,101]. In amoeba, fungi, and animals, proper cytokinesis requires the assembly of an NM2-enriched actomyosin contractile ring, which pulls the plasma membrane to the cell equator to physically separate the daughter cells [102]. The NM2 activity is also required for cellular polarization and further asymmetrical cell division [103,104]. Although NM2 is critical for cytokinesis, its motor activity is dispensable in budding yeast. In animals, the requirement of ATPase activity is still debatable. Some studies suggest that NM2 motor activity is not necessary for cytokinesis [98,105,106,107] and any of the three paralogs would promote a successful division of the cell [108]. In contrast, specific mutations affecting the ATPase activity of NMY-2 in *C. elegans* show that its motor activity, rather than its cross-linking activity on actin filaments, is required for cytokinesis [91].

*Adhesion:* Cell adhesion to the extracellular matrix (ECM) relies on integrins, which together with actin filaments and NM2-produced tension promote cell migration [109]. Considering that among the NM2 isoforms NM2A is the fastest, it has the highest actin-activated ATPase activity and the highest rate of actin filaments sliding [50], its function fits better with rapid actin remodeling and assembly of focal adhesions at the leading edge of migrating cells [54,110,111].

Intercellular adhesion contacts are constituted by the apical and basolateral junctional complexes that define cell polarity [112]. Junction complexes assemble, mature and disassemble at the apical zone of polarized cells, where junctional integrity and stabilization are preserved by their connection with actomyosin bundles [113]. Blebbistatin-inhibition of NM2 activity decreases both the size and the density of adhesion complexes [114], which demonstrates the importance of NM2 activity on the proper dynamics of such structures. NM2 contractile forces are fundamental to power the contraction of the adherens junctions both in vitro and in vivo [115,116], having a key role in the establishment of polarized epithelial tissues [117]. In particular, NM2A contractile activity is required for the assembly, maturation and temporal stability of adhesion complexes [12,118,119]. In intestinal epithelial cells, depletion of NM2A by siRNA results in the disruption of cell–cell adhesion and induces profound alterations on cell morphology [120]. In kidney epithelial cells, NM2A knockdown shortens adhesion contacts and disrupts stable intercellular adhesions [119]. In line, NM2A ablation in mice decreased the levels of E-cadherin and β-catenin at the adhesion sites leading to the loss of cell–cell contacts [89].

*Cell migration:* To migrate in 2D environments, cells disrupt their symmetry and polarize into a leading edge and a cell rear, which requires actomyosin cytoskeleton reorganization [121,122]. NM2 isoforms play important roles in 2D migration and their inhibition correlates with impaired cell motility and decreased speed [123,124,125]. Polymerization of actin filaments at the leading edge promotes lamellipodium extension, coupled to an NM2-dependent actin retrograde flow that enhances the assembly of novel focal adhesions at the cell front [126,127,128,129,130]. While NM2A preferentially accumulates at the leading edge, NM2B concentrates at the rear (Figure 6A). The current mechanism underlying such segregation proposes that throughout the retrograde flow of actin filaments, NM2A is more prominent in nascent filaments due to its faster rate of disassembly from heterotypic NM2A/NM2B bipolar filaments, whereas NM2B remains associated with old actin fibers, accumulating at the cell rear [54]. Despite the well-established NM2A/NM2B segregation, NM2A is also essential for the translocation of the cell body and for the disassembly of mature adhesions at the rear, thus assisting retraction of the trailing edge and forward movement [131,132,133,134]. In addition, while NM2A is associated with cell contraction and promotes directional cell migration, excessive levels of NM2B suppress cell motility and may stabilize the cytoskeleton [54]. The expression levels of the different isoforms may thus dictate the migratory behavior of cells in various environments.

In 3D environments, migration is classified in several types, mesenchymal and ameboid migratory behaviors being the most common [135]. They differ in the degree of adhesion to the ECM, NM2-powered traction forces and NM2 cellular localization [135]. The mesenchymal mode shares similarities with 2D migration. It is characterized by adhesions to the ECM and the formation of leading edge protrusions enriched in NM2 and actin filaments such as lamellipodia, filopodia or actin spikes [136]. Cells moving on 3D environments are elongated and the protrusions are thinner. In the amoeboid mode of migration, ECM adhesions are dispensable and NM2 is either localized at the front or rear of the cell [137,138]. Cells display more actomyosin contractile forces that promote the formation of plasma membrane protrusions (blebs) at the leading edge [139], allowing the cells to move forward. Amoeboid migration is also associated with leukocyte movement, which requires the assembly of highly contractile structures at the rear, called uropods [137,140]. The contractile forces generated by Rho, ROCK and NM2 disrupt adhesions at the rear, allowing cells to move [141,142].

## 9. NM2A in Membrane Blebbing

Plasma membrane blebbing occurs both in physiological and pathological conditions and was reported during cell migration, cell division and apoptosis [143,144,145]. Blebbing is also a general cellular response to plasma membrane injury, correlated to bacterial infections and intoxication with bacterial pore-forming toxins [146]. Blebs are dynamic protrusions regulated by different cytoskeletal molecules [147,148]. They result from the contraction of the cortical actomyosin cytoskeleton, which creates intracellular pressure [147,148,149,150] that weakens the interaction between the plasma membrane and the underlying cytoskeleton, thus allowing the membrane to protrude [149,151,152,153,154]. The continuous hydrostatic pressure inside cells further promotes bleb expansion [155], which will eventually retract slowly (Figure 6B). Bleb expansion and retraction mostly depend on the actomyosin cytoskeleton [151]. Expanded blebs lack polymerized actin or other cytoskeletal elements [156]. Before retraction, plasma membrane-actin connector proteins at the bleb (e.g., ezrin) recruit actin and actin nucleators that initiate actin polymerization [151]. NM2A is further recruited, accumulates on actin filaments and powers the retraction of the bleb [139,145,157,158] (Figure 6B).

At the molecular level, the GTPase RhoA and its downstream effector kinase ROCK promote the activation of NM2 and drive actomyosin contraction [145,153,156,157]. RhoA is activated by MYOGEF, which is recruited by ezrin to the bleb [157]. Inhibitors of ROCK and MLCK, as well as blebbistatin and actin-depolymerizing drugs repress membrane blebbing [149,159]. Despite the fact that all NM2 paralogs may be recruited to the blebs, NM2A is the one driving bleb retraction in cells [160]. While the NM2A motor activity is necessary and sufficient to regulate bleb retraction, its non-helical tail is also necessary but not essential [160]. Moreover, NM2A turnover on bipolar filaments correlates with the blebbing retraction rate [144,160,161]. Interestingly, calcium chelators inhibit blebbing [145], suggesting that calcium is required for NM2 contraction and subsequent bleb formation and retraction. Indeed, blebbing is triggered by a variety of calcium-dependent mechanisms known to activate NM2 and/or induce cytoskeleton rearrangements such as plasma membrane repair upon bacterial-induced pore formation [146,162,163,164,165,166].

## 10. NM2A in Disease

NM2A-associated diseases (Figure 7) either result from mutations on the heavy chains, alternative splicing errors or misregulation of the NM2A activity [167,168]. They affect the vascular and nervous system, and different organs such as the kidneys. Given that the heavy chain carries the majority of the mutations, these illnesses are known as *MYH9*-related diseases (*MYH9*-RD), autosomal dominant disorders that result in a plethora of syndromes [169,170,171,172]. The molecular basis linking *MYH9*-RD to the function of NM2A was recently reviewed [173]. The most prevalent mutation occurs on Arginine 702 (R702) located in the motor domain. This mutation reduces myosin ATPase activity and was associated with decreased circulatory platelets (thrombocytopenia), nephritis and deafness [174]. *MYH9*-RD were also associated with splicing, nonsense and frameshift mutations on the NMHC2A non-helical tailpiece, as well as to in-frame deletions or duplications in repetitive-sequence regions [175,176,177,178]. *MYH9*-RD-affected patients display thrombocytopenia, platelet macrocytosis and NMHC2A aggregation in neutrophil granulocytes [179,180]. Differentiation and maturation of the platelet precursors (megakaryocytes) are dependent on NM2A, which is the only NM2 isoform expressed in platelets [181,182,183,184]. In addition, patients may develop deafness, kidney inflammation, cataracts and abnormal enzymatic activity on the liver [174,185,186].

Recent evidences suggest that abnormal activity of NM2 is involved in neurological diseases. Interestingly, *MYH9* is among the three genes affected simultaneously in schizophrenia, intellectual disability and autism [187]. In addition, increased levels of phosphorylated RLCs were found in brain tissues of schizophrenic individuals, and miRNAs associated with NM2 regulation are either upregulated during amyotrophic lateral sclerosis (ALS) or promote inflammation during multiple sclerosis [188,189,190]. In line, inhibitors of ROCK are being tested for treatment of drug addiction and neurodegenerative diseases such as Parkinson’s and Alzheimer’s [191,192,193,194] and vasodilator drugs targeting NM2 have been administered in Alzheimer’s, ALS and Parkinson’s mouse models [195,196,197,198].

Due to its function in both cell adhesion and migration, NM2 misregulation or malfunction was associated with cancer and atherosclerosis. Inhibition of MLCK or ROCK was shown to reduce atherosclerosis in mice and blebbistatin diminishes kidney inflammation in vivo by impairing leukocyte infiltration [199,200]. Altered NM2 expression and/or activation affect cell migration and cell division, which are hallmarks of tumor growth and invasion [201,202]. NM2 is directly or indirectly regulated by many oncogenes, miRNAs, cytokines and tumor microenvironment-associated adhesion forces. Often, tumor migratory and invasive behaviors are NM2-dependent and occur through ROCK and MLCK activation. In vitro, the inhibition of NM2 activity by blebbistatin prevents the migration and invasion of breast cancer and glioma cells [203,204]. NM2 has thus emerged as a potential target for the treatment of tumors with high invasive capacity. Despite that the upregulation of NM2A was associated with tumorigenesis and cancer invasion [205,206,207,208], NM2A is also considered a tumor suppressor. In fact, NM2A protects skin cells from uncontrolled rounds of cell division and NM2A-associated mutations promote skin cancer development [209]. In addition, low levels of NM2A mRNA were linked with poor prognosis in patients with head and neck squamous cell carcinoma [209].

NM2A was also linked to viral and bacterial infectious diseases: Its depletion often impairs the dissemination of pathogens [210]. Infection studies demonstrated that pathogens such as Kaposi’s sarcoma-associated herpesvirus and *Listeria monocytogenes* target NM2A, inducing its phosphorylation, which is detrimental for cellular invasion [75,211]. Listeriolysin O (LLO), a pore-forming toxin produced by *L. monocytogenes*, induces the redistribution and remodeling of NM2A into cortical bundles that promote plasma membrane (PM) repair after LLO-induced PM disruption [162,163]. This mechanism is shared by other pore-forming toxins, such as PFO that is expressed by *Clostridium perfringens* [166], suggesting a protective role for NM2A during infection by bacterial pathogens.

## 11. Concluding Remarks

NM2A is a central protein in cell mechanics, which plays key roles during mechanosensing and as an intracellular force generator, thus assisting a variety of essential cell biology processes. Our current knowledge reveals that regulation of NM2A is multifaceted and that a network of factors allows myosin to differently function in diverse cellular contexts and in response to a variety of stimuli. New findings describing novel post-translational modifications and mixed bipolar filaments suggest the existence of a very complex regulatory network that contributes to the fine regulation of NM2A activity as well as to its cellular localization. Efforts should now go to the identification of the mechanisms controlling NM2A in space and time at the cellular and tissue level.

Due to its direct or indirect contribution to several diseases, NM2A emerged as a suitable target for future therapeutic approaches to treat a variety of diseases. However, the pleiotropic roles of NM2A in cells invalidate the use of drugs that would indiscriminately target the total pool of NM2A in cells. The understanding of the mechanisms governing specific cellular localization and underlying the co-polymerization of mixed myosin filaments should provide new candidates to target specific pools of NM2A. 

## Figures and Tables

**Figure 1 cells-09-01590-f001:**
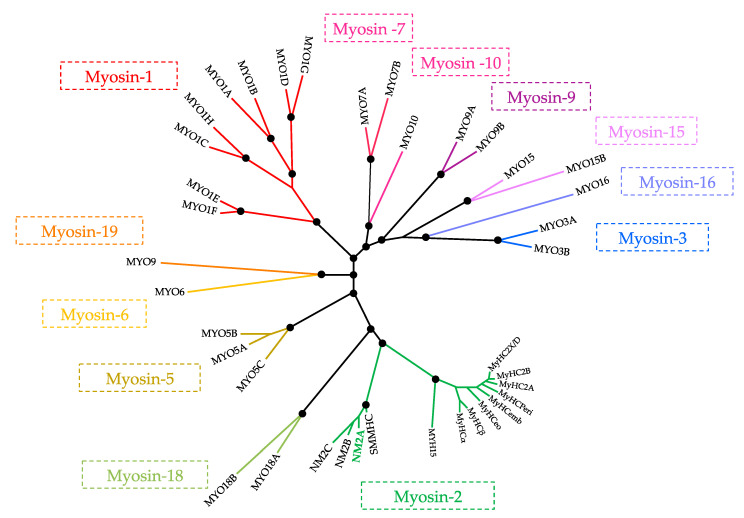
Phylogenetic tree of the myosin superfamily in humans. The 38 human genes that encode the diverse myosins are identified. The sequences of the myosin motor domains were analyzed and grouped phylogenetically allowing the distribution of myosins into 12 different classes. This figure was adapted from [8].

**Figure 2 cells-09-01590-f002:**
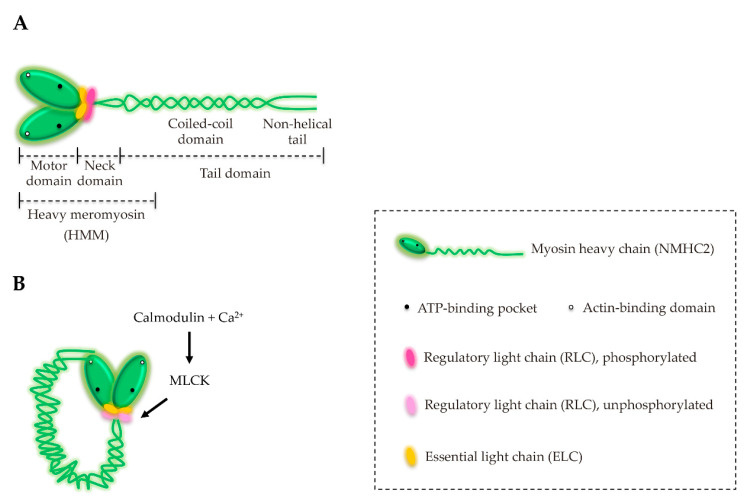
The structure of non-muscle myosin 2 (NM2). (**A**) NM2 is a hexamer composed by two heavy chains (green) and four light chains: Two regulatory (RLC, pink) and two essential light chains (ELC, yellow). The full NM2 complex can be structurally divided in three regions: The motor, the neck and the tail domains. The head domain includes the ATP- and actin-binding sites. The light chains associate with the heavy chains in the neck domain. The heavy meromyosin (HMM) is an NM2 fragment obtained by tryptic digestion and commonly used in in vitro biochemical assays. Even lacking part of the tail, HMM maintains the structure and the biochemical properties of the full molecule. (**B**) Schematic representation of the inactive NM2 molecule. Unphosphorylated RLC (light pink) promotes the intramolecular interaction between the tail and the motor domain. The phosphorylation of RLC Serine 19 by Myosin light chain kinase (MLCK) turns on NM2, promoting the assembly of filaments and increasing ATPase activity.

**Figure 3 cells-09-01590-f003:**
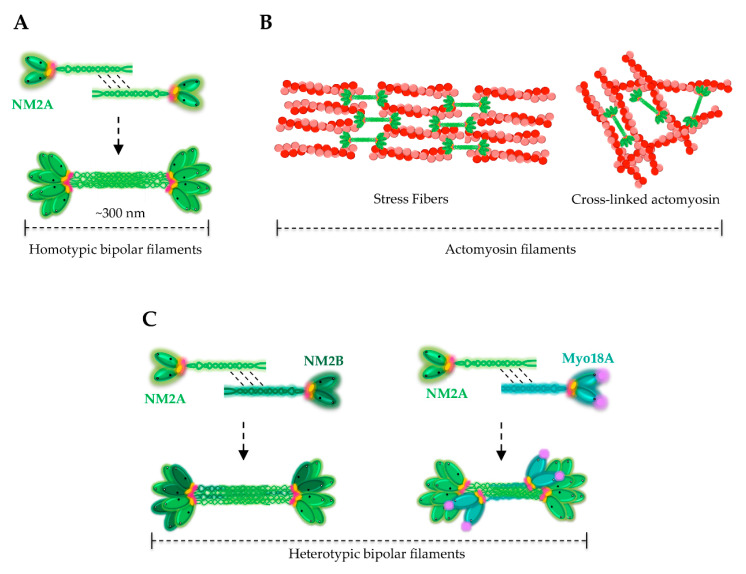
Mechanism of NM2 assembly and binding to actin filaments. (**A**) Assembly of homotypic bipolar filaments of NM2A. NM2A molecules interact antiparallelly by their tail regions and assemble into NM2A bipolar filaments of around 300 nm in length. The NM2A motor domains are oriented to the outside of the polymer and are free to interact with polymerized actin. (**B**) NM2A polymers bind to actin filaments building up stress fibers or more dynamic cross-linked actomyosin meshworks. (**C**) Assembly of heterotypic bipolar filaments. Different myosins are able to co-polymerize originating mixed filaments which may have different kinetic properties. Extra domains of Myo18A (purple, PDZ domain) may allow the interaction with additional proteins possibly increasing the layers of NM2 regulation.

**Figure 4 cells-09-01590-f004:**
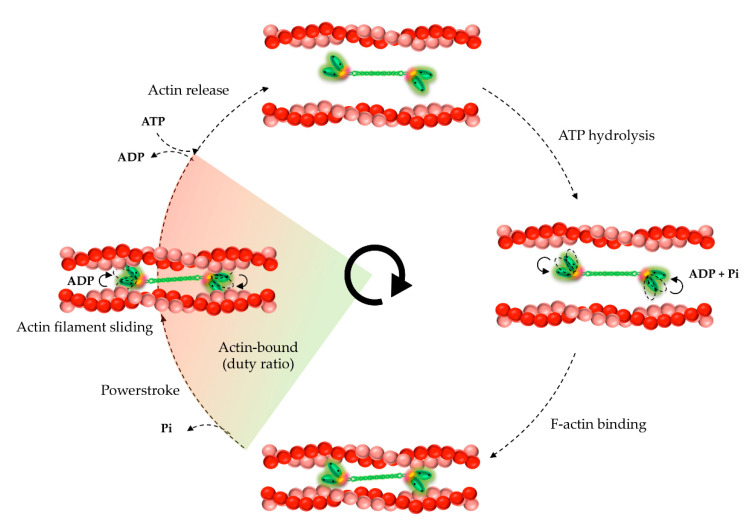
Schematic model for NM2-dependent actin sliding cycle. ATP binding dissociates NM2 from actin (actin release). Myosin motor head hydrolyzes ATP and re-attaches to actin. The release of the phosphate (Pi) triggers a conformational change that promotes movement and powers the contraction of actin filaments (powerstroke). The colored region corresponds to the portion of the cycle during which the NM2 motor head is strongly bound to actin, this is called duty ratio. The exchange of ADP by ATP restarts the cycle. This figure was adapted from [65].

**Figure 5 cells-09-01590-f005:**
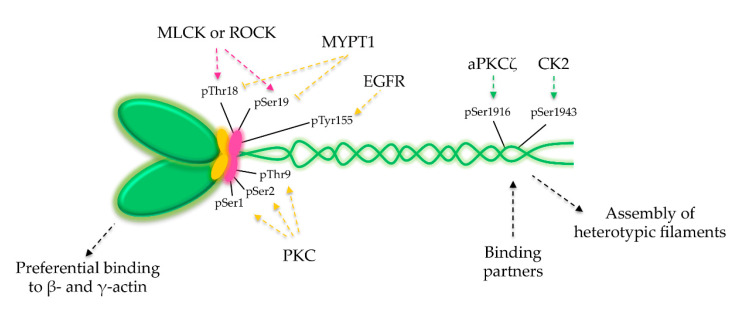
Multiple mechanisms regulate NM2A activity and subcellular localization. NM2A activation (pink arrows) and inactivation (yellow arrows) are mainly regulated through RLC phosphorylation or dephosphorylation on Ser, Thr and Tyr residues. Events occurring at the NM2A tail, such as phosphorylations (green arrows), interaction with binding partners and co-polymerization of heterotypic filaments, control the assembly and disassembly of NM2 filaments. Regulation through the head domain is determined by the differential binding of NMHC2 to distinct actin isoforms.

**Figure 6 cells-09-01590-f006:**
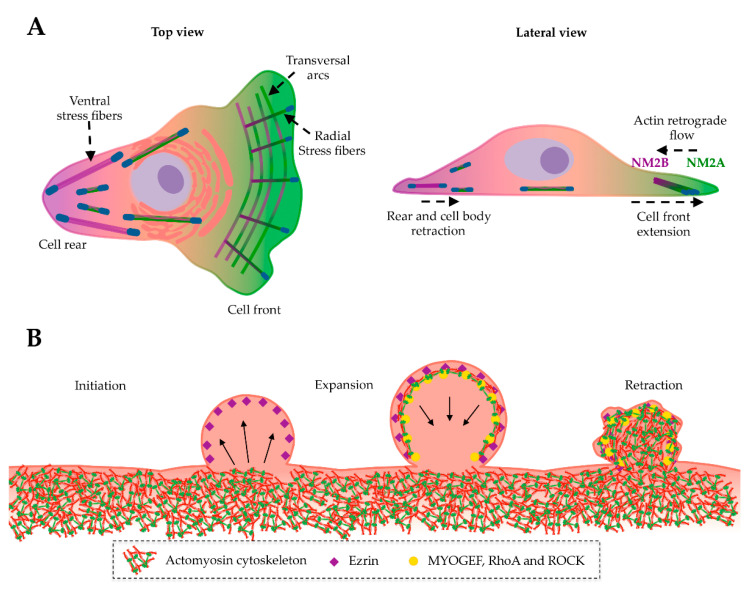
Involvement of NM2A in cell migration and plasma membrane blebbing. (**A**) Scheme showing a cell migrating in a 2D environment, top and lateral views are represented. To migrate in a 2D environment the cell polarizes in the front and rear regions, which are defined by an intracellular gradient of proteins and allow directionality during movement. NM2 isoforms play a key role in 2D migration and are selectively localized at different cell regions. NM2A concentrates at the cell front where high actin dynamics are taking place, and gradually decreases towards the rear where NM2B is concentrated. In between cell poles (front and rear) NM2A and NM2B co-exist at different concentrations in mixed actomyosin filaments. Focal adhesions at the extremities of stress fibers are shown as blue circles. (**B**) Process of plasma membrane bleb formation and retraction. Upon stimuli (e.g., apoptosis and calcium influx), a membrane protrusion forms (bleb initiation) due to the plasma membrane detachment from the cortical cytoskeleton. As a result of intracellular pressure, the bleb swells (bleb expansion) until ezrin recruits cytoskeletal proteins that promote actin polymerization and the subsequent interaction with NM2A. NM2A contractile forces pull the membrane inwards and support bleb retraction.

**Figure 7 cells-09-01590-f007:**
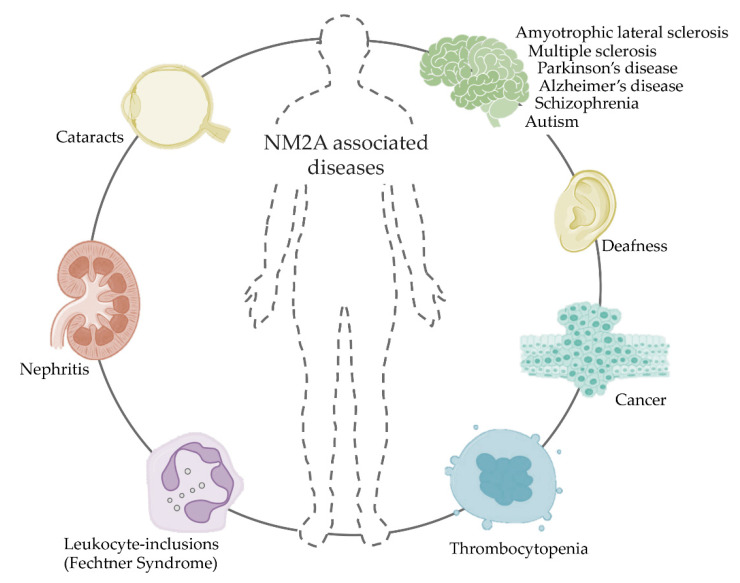
Human pathologies associated with mutations in *MYH9*, NM2A expression and/or activity defects. In agreement with the key roles of NM2A from embryo development to adulthood, its malfunction affects several organs causing mild to life-threatening disorders. In particular, NM2A defects were associated with neurodegenerative diseases, cancer, bleeding diseases and kidney inflammation.

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
