# Peer review of "Non-Muscle Myosin 2A (NM2A): Structure, Regulation and Function"

_cells, 2020, doi:10.3390/cells9071590_

Round 1

Reviewer 1 Report

Dear Authors,

This review is timely. You have tried to address the structure, regulation and function of NM2A but due to the limitation in word count non of the sections are detailed enough.

1) Please get a native English speaker to edit the manuscript. The manuscript has too many grammatical errors for me to list them down.

2) The figure legends are very  poor or hardly any eg Fig 8.

3) The text within the figures are not clear.

4) Line 307: NMIIA, same as NM2A? Please be consistent

Author Response

REFEREE 1

We are grateful to the reviewer for his/her comments. We have addressed all the issues raised and proposed a new improved version of our manuscript. A point-by-point response is provided below.

Point-by-point response to the reviewer’s comments:

1) Please get a native English speaker to edit the manuscript. The manuscript has too many grammatical errors for me to list them down.

Authors response: Extensive editing of English language was performed in the new version of our manuscript.

2) The figure legends are very poor or hardly any eg Fig 8.

Authors response: All the figure legends were reviewed and completed to include detailed description of the proposed schemes.

3) The text within the figures are not clear.

Authors response: An unexpected problem occurred in all the figures in the PDF document provided to the reviewers. We apologize for that. We have done our best to avoid further problems in our revised version of the manuscript.

4) Line 307: NMIIA, same as NM2A? Please be consistent

Authors response: Corrected in the new version of the manuscript.

Reviewer 2 Report

The manuscript reviewing non-muscle myosin 2A provides a concise overview of this important motor protein. A number of points though need to be addressed which would strengthen the review.

(1) Line 29: Should also include mention of muscle contraction

(2) Line 37: How do the 4 classes noted here differ from classes noted in the previous paragraph?

(3) Figures 2-8 have a font type where characters appear to be not evenly spaced.

(4) Line 56: reference is made to the presence of non-muscle myosin in sarcomeric structures of skeletal muscle. I presume the authors are referring to muscle myosin? It would be worthwhile to point out NM is mainly present in stress fibres in non-muscle cells at this point as well.

(5) In the same context figure 4 shows NM2 in both sarcomeric structures and myosin stacks which appear to be the same. In reality these are quite different due to presence of anchoring/stabilising and spacing proteins which are not mentioned. Again, is NM2 present in sarcomeres of skeletal muscle?

(6) Line 61: would be polypeptides or proteins not peptides.

(7) Line 101: Sentence needs to be corrected to “interact with it.”

(8) Line 103: can the authors elaborate further on why would only certain LCs interact with certain HCs?

(9) Line 119: Again, still would mainly be in the context of stress fibres?

(10) With reference to figure 3 and regulation of NM2 it would be worth highlighting differences in regulation to that of skeletal muscle myosin. No mention of calcium/calmodulin/MLCK pathway is made in the text although it is shown in the figure 3. MLCK should be defined in figure 3.

(11) Preferential binding to actin isoforms is highlighted in figure 7 but the concept of actin isoforms and relationship to myosin is not discussed.

(12) Line 229: What binding partners?

(13) Some concluding remarks should be added summarising the review and what is still needed to be explored regarding the role and regulation of NM2.

Author Response

REFEREE 2

We are grateful to the reviewer for his/her comments. We have addressed all the issues raised and proposed a new improved version of our manuscript. In particular, we added a “Concluding remarks” section. A point-by-point response is provided below.

Point-by-point response to the reviewer’s comments:

(1) Line 29: Should also include mention of muscle contraction

Authors response: Included in the new version of our manuscript (line 40).

(2) Line 37: How do the 4 classes noted here differ from classes noted in the previous paragraph?

Authors response: The differences rely on the use of different parameters to classify myosins. The mentioned 12 classes classify myosins according to their evolutionary and structural differences. Myosins can still be divided in functional classes that, as we stated, vary regarding the ATPase activity, time pent attached to actin (duty ratio) and velocity with which they slide on or move along actin. For instance, myosins from class 5 and 6 belong to the class of fast movers. To clarify we replaced “classes” by “groups” in line 52.

(3) Figures 2-8 have a font type where characters appear to be not evenly spaced.

Authors response: An unexpected problem occurred in all the figures in the PDF document provided to the reviewers. We apologize for that. We have done our best to avoid further problems in our revised version of the manuscript.

(4) Line 56: reference is made to the presence of non-muscle myosin in sarcomeric structures of skeletal muscle. I presume the authors are referring to muscle myosin? It would be worthwhile to point out NM is mainly present in stress fibres in non-muscle cells at this point as well.

Authors response: Modified in the new version of the manuscript (lines 69-71).

(5) In the same context figure 4 shows NM2 in both sarcomeric structures and myosin stacks which appear to be the same. In reality these are quite different due to presence of anchoring/stabilising and spacing proteins which are not mentioned. Again, is NM2 present in sarcomeres of skeletal muscle?

Authors response: We are sorry for the ambiguity. We modified Fig 4 and provide a revised version. Indeed, NM2 is not present in sarcomeres of skeletal muscle.

(6) Line 61: would be polypeptides or proteins not peptides.

Authors response: We agree with the referee. This was modified in the new version of our manuscript (line 78).

(7) Line 101: Sentence needs to be corrected to “interact with it.”

Authors response: This section is re-structured in the new version of the manuscript (lines 117-125).

(8) Line 103: can the authors elaborate further on why would only certain LCs interact with certain HCs?

Authors response: The molecular basis for specific interactions between LCs and HCs are unknown. It is plausible that the expression levels in certain tissues or cellular localization favor specific interactions. The paragraph mentioning specific LCs and HCs interactions was modified in the new version of the manuscript (lines 115-123).

(9) Line 119: Again, still would mainly be in the context of stress fibres?

Authors response: Altered in the new version of the manuscript (lines 137-138).

(10) With reference to figure 3 and regulation of NM2 it would be worth highlighting differences in regulation to that of skeletal muscle myosin. No mention of calcium/calmodulin/MLCK pathway is made in the text although it is shown in the figure 3. MLCK should be defined in figure 3.

Authors response: Figure 3 was modified, now figure 2B. Calcium/calmodulin/MLCK pathway is now mentioned in the text (lines 132-133).

(11) Preferential binding to actin isoforms is highlighted in figure 7 but the concept of actin isoforms and relationship to myosin is not discussed.

Authors response: Added in the new version of the manuscript (lines 310-311).

(12) Line 229: What binding partners?

Authors response: Binding partners are mentioned in the new version of the manuscript (lines 291-292, 303-306).

(13) Some concluding remarks should be added summarising the review and what is still needed to be explored regarding the role and regulation of NM2.

Authors response: In agreement with the referee’s request, the new version of our manuscript includes a new section entitled “concluding remarks”.

Reviewer 3 Report

This manuscript well summarized current research findings on NM2 structure and function as well as its association with a spectrum of disorders. The manuscript contains 8 figures and the first 7 figures are covering the basics of myosin 2 and the equivalent information can be found elsewhere. Therefore, at least few of them can be deleted and few new diagrams depicting NM2 functions in development, cell migration, and membrane blebbing would be more beneficial to the readers. In addition, this manuscript may need a conclusion section or a concluding remark section that illustrate potential problems that need to be addressed in the near future, which can provide some insights on new research trends in this field. Overall, the manuscript has quality of information that warrants it acceptance to the journal, but the following changes are need to be made along with new figures (as stated above) and a concluding remark section.

Page 1

Lines 21-23: since microfilaments and microtubules are serving as highways for the indicated motors, the current sentence needs to be appropriately revised.

Page 2

Lines 61-63: the sentence is not clear in explaining the composition of the hexamer. Revision is required.

Page 3

Figure 2 text or labeling is not consistent with the font style used in other figure such as Figure 1. Further, all words in Figure 2 are connected without space. Revision is required.

Page 4

Figure 3: the font style of “calmodulin” and “MLCK” can be consistent with other labeled words.

Page 5

Figure 4: font style needs to be consistent with figure 1. Fix typos such as “filment”, “filmenta”. In addition, in non-muscle cells, myosin is a component of the stress fiber and so instead of myosin stacks, it is better label it as stress fiber.

Figure 5: Font style to be revised. In addition, the figure illustrates PDZ domain. However, the main text does not cite the function of the domain neither in the figure caption.

Page 6

Figure 6: font style to be revised. Fix typos.

Page 7

Line 200: primaryè primarily

Figure 7: label font to be consistent with all other figures.

Page 8

Line 232: tailor? Isn’t this a typo?

Page 9

Line 294: NM2A siRNA-depletionè this phrase needs to be changed into something like “ NM2A depletion via siRNA….

Line 307: NMIIAè NM2A?

Page 11

Figure 8: font style consistency.

Line 388: NM2 plays a role in disorders è normal version of it does not play a role. Therefore, the phrase needs to be revised.

Similarly, Line 405: NM2A has been also involved in either viral or bacterial infectious diseases:è May be it needs to be stated like “ alteration of NM2A levels…”

Page 12

This manuscript may need a conclusion section or a concluding remark section that illustrate potential problems that need to be addressed in the near future, which can provide some insights on new research trends in this field.

Author Response

REFEREE 3

We are grateful to the reviewer for his/her comments. We have addressed all the issues raised and proposed a new improved version of our manuscript.

Concerning the figures, we understand the point of the reviewer. Indeed, our figures mainly cover the basics of NM2 and despite equivalent schemes can be found elsewhere we think that it is easier to the reader to maintain them in our manuscript. We slightly altered our figures in the new version of the manuscript. In addition, we included, as suggested by the referee, a new figure (Figure 6) focusing on the role of NM2A in cell migration and plasma membrane blebbing.

Our new version of the manuscript also includes a new section entitled “Concluding remarks” as suggested by the referee.

A point-by-point response is provided below.

Point-by-point response to the reviewer’s comments:

Page 1

Lines 21-23: since microfilaments and microtubules are serving as highways for the indicated motors, the current sentence needs to be appropriately revised.

Authors response: Modified in the new version of the manuscript (lines 26-31).

Page 2

Lines 61-63: the sentence is not clear in explaining the composition of the hexamer. Revision is required.

Authors response: Modified in the new version of the manuscript (line523 )

Page 3

Figure 2 text or labeling is not consistent with the font style used in other figure such as Figure 1. Further, all words in Figure 2 are connected without space. Revision is required.

Authors response: An unexpected problem occurred in all the figures in the PDF document provided to the reviewers. We apologize about that. We have done our best to avoid this kind of problem in our revised version of the manuscript.

Page 4

Figure 3: the font style of “calmodulin” and “MLCK” can be consistent with other labeled words.

Authors response: An unexpected problem occurred in all the figures in the PDF document provided to the reviewers. We apologize about that. We have done our best to avoid this kind of problem in our revised version of the manuscript.

Page 5

Figure 4: font style needs to be consistent with figure 1. Fix typos such as “filment”, “filmenta”. In addition, in non-muscle cells, myosin is a component of the stress fiber and so instead of myosin stacks, it is better label it as stress fiber.

Authors response: An unexpected problem occurred in all the figures in the PDF document provided to the reviewers. We apologize about that. We have done our best to avoid this kind of problem in our revised version of the manuscript. Typos were corrected and modifications were done according to the suggestions of the referee.

Figure 5: Font style to be revised. In addition, the figure illustrates PDZ domain. However, the main text does not cite the function of the domain neither in the figure caption.

Authors response: An unexpected problem occurred in all the figures in the PDF document provided to the reviewers. We apologize about that. We have done our best to avoid this kind of problem in our revised version of the manuscript. PDZ domain harbored by Myo18A is now mentioned in the Figure caption as well as in the text (lines 177-179).

Page 6

Figure 6: font style to be revised. Fix typos.

Authors response: Modified in the new version of the manuscript.

Page 7

Line 200: primaryè primarily

Authors response: Corrected in the new version of the manuscript (line 255)

Figure 7: label font to be consistent with all other figures.

Authors response: Modified in the new version of the manuscript.

Page 8

Line 232: tailor? Isn’t this a typo?

Authors response: Modified in the new version of the manuscript.

Page 9

Line 294: NM2A siRNA-depletionè this phrase needs to be changed into something like “ NM2A depletion via siRNA….

Authors response: Modified in the new version of the manuscript (line 382)

Line 307: NMIIAè NM2A?

Authors response: Corrected in the new version of the manuscript (line 396)

Page 11

Figure 8: font style consistency.

Authors response: Modified in the new version of the manuscript.

Line 388: NM2 plays a role in disorders è normal version of it does not play a role. Therefore, the phrase needs to be revised.

Authors response: Modified in the new version of the manuscript (lines 499-500).

Similarly, Line 405: NM2A has been also involved in either viral or bacterial infectious diseases:è May be it needs to be stated like “ alteration of NM2A levels…”

Authors response: Modified in the new version of the manuscript (line 523).

Page 12

This manuscript may need a conclusion section or a concluding remark section that illustrate potential problems that need to be addressed in the near future, which can provide some insights on new research trends in this field.

Authors response: In agreement with the referee’s request, the new version of our manuscript includes a new section entitled “concluding remarks”.

Round 2

Reviewer 1 Report

The revised manuscript is much better and easier to read but the English needs to be improved further.

Eg. line 133: On their active conformation, ....

It should be in their active conformation, ...

Eg. line 140: NM2 bipolar filaments have been long considered as homotypic polymers constituted exclusively by one of the NM2 isoforms.

It should be: NM2 bipolar filaments have long been considered as homotypic polymers. 

Reviewer 2 Report

The manuscript has been significantly strengthened in response to reviewer's comments.